# Graphene Far-Infrared Irradiation Can Effectively Relieve the Blood Pressure Level of Rat Untr-HT in Primary Hypertension

**DOI:** 10.3390/ijms25126675

**Published:** 2024-06-18

**Authors:** Guanjie Lu, Haotong Guo, Yi Zhang, Meng Zhang, Tao Zhang, Ge Hu, Qian Zhang

**Affiliations:** College of Animal Science and Technology, Beijing University of Agriculture, Beijing 102206, China; luguanjie2023@163.com (G.L.); bedavzyl@163.com (H.G.); 15024928657@163.com (Y.Z.); 18769011246@163.com (M.Z.); zhangtao@bua.edu.cn (T.Z.); bnhuge@126.com (G.H.)

**Keywords:** graphene far-infrared irradiation, vascular smooth muscle, primary hypertension

## Abstract

Graphene, when electrified, generates far-infrared radiation within the wavelength range of 4 μm to 14 μm. This range closely aligns with the far-infrared band (3 μm to 15 μm), which produces unique physiological effects. Contraction and relaxation of vascular smooth muscle play a significant role in primary hypertension, involving the nitric oxide-soluble guanylate cyclase–cyclic guanosine monophosphate pathway and the renin–angiotensin–aldosterone system. This study utilized spontaneously hypertensive rats (SHRs) as an untr-HT to investigate the impact of far-infrared radiation at specific wavelengths generated by electrified graphene on vascular smooth muscle and blood pressure. After 7 weeks, the blood pressure of the untr-HT group rats decreased significantly with a notable reduction in the number of vascular wall cells and the thickness of the vascular wall, as well as a decreased ratio of vessel wall thickness to lumen diameter. Additionally, blood flow perfusion significantly increased, and the expression of F-actin in vascular smooth muscle myosin decreased significantly. Serum levels of angiotensin II (Ang-II) and endothelin 1 (ET-1) were significantly reduced, while nitric oxide synthase (eNOS) expression increased significantly. At the protein level, eNOS expression decreased significantly, while α-SMA expression increased significantly in aortic tissue. At the gene level, expressions of *eNOS* and *α-SMA* in aortic tissue significantly increased. Furthermore, the content of nitric oxide (NO) in the SHR’s aortic tissue increased significantly. These findings confirm that graphene far-infrared radiation enhances microcirculation, regulates cytokines affecting vascular smooth muscle contraction, and modifies vascular morphology and smooth muscle phenotype, offering relief for primary hypertension.

## 1. Introduction

Graphene, a new type of graphite material, was first reported in 2004 [1]. It is a two-dimensional carbon nanomaterial consisting of a hexagonal honeycomb structure composed of single-layer carbon atoms with sp2 hybrid orbitals, and its thickness is only 0.35 nm [2]. Its unique structure makes it widely used in many fields such as electronics, optics, mechanics, and thermodynamics [3,4]. In recent years, an increasing number of researchers have found that graphene, as a material, also has great potential in the field of biomedicine. As living organisms are natural absorbers of far-infrared light, with the absorbed wavelength mainly between 3 μm and 15 μm [5], the wavelength of far-infrared radiation emitted by graphene material after being electrically stimulated is between 4 μm and 14 μm, which is close to the infrared wavelength range that is easily absorbed by living organisms [5]. Moreover, studies have shown that the infrared spectrum emitted by graphene after being electrified has spectral features that are very similar to the infrared spectrum of the human body. Far-infrared radiation can penetrate 5 mm deep into the subcutaneous tissue of the body, enhance microvascular contractility, promote blood circulation, and regulate blood pressure, improving functional disorders caused by organ ischemia such as the heart, liver, and kidneys [6]. In addition, it can improve endothelial cell function, reduce the incidence of certain vascular-related diseases [7], and support the growth of various cells such as fibroblasts, bone marrow mesenchymal stem cells, and neural stem cells [8]. Researchers have reported the potential use of graphene material emitting far-infrared radiation after being electrified in tumor treatment [9]. Meanwhile, studies have also shown that graphene material may have a good effect on dental care [10] and wound healing [11]. In addition, in 2018, graphene far-infrared irradiation was certified by the Food and Drug Administration (FDA) to have safe biological effects (Regulation Number: 890.5740).

Hypertension is a significant risk factor for cardiovascular diseases (CVDs) and one of the leading causes of premature death worldwide [12]. Due to factors such as aging populations and changes in lifestyle, the global incidence of hypertension is increasing [13]. According to current research, approximately 30% of the global adult population suffered from hypertension between 1975 and 2015 [12], and it is estimated that by 2025, there will be 1.5 billion adults with hypertension worldwide [14] due to factors such as the aging population and lifestyle risks. The decrease in vascular contractility and the increase in intravascular fluid volume are two direct causes of primary hypertension, and the factors that can lead to these two direct causes are potential causes of primary hypertension. These factors are often interdependent and intertwined in the development of primary hypertension [15]. Although the use of various drugs in clinical practice can lower blood pressure levels in patients, it can also potentially cause side effects to patients. Therefore, we are attempting to identify effective non-pharmacological treatment methods. Based on the background mentioned above, it can be inferred that graphene far-infrared radiation may have a certain relieving effect on the blood pressure level of patients with primary hypertension. Currently, there is no research on the effect of this material on primary hypertension. Therefore, in this study, we used the SHR untr-HT to explore the effects of graphene far-infrared radiation on blood pressure-related indicators and the structure of vascular endothelial cells and vascular smooth muscle. The objective of this study is to provide a theoretical basis for the development of products and clinical applications of graphene as a potential non-pharmacological treatment option for primary hypertension.

## 2. Results

### 2.1. The Improvement in Tail Artery Blood Pressure and Blood Flow Perfusion in the SHR Untr-HT by Graphene Far-Infrared Irradiation

Changes in blood pressure and blood flow perfusion were recorded for each group of rats over a period of 7 weeks. As can be seen from the figure, rats in the Nifedipin group and Gra-I group exhibited an overall decreasing trend in systolic pressure (Figure 1A) and diastolic pressure (Figure 1B), while the systolic and diastolic pressure in the Control and untr-HT groups remained relatively stable.

Figure 1C shows the changes in microcirculation blood flow perfusion of rats in each group during the 7-week period. As can be seen from the graph, the microcirculation blood flow perfusion of the Gra-I group rats was significantly higher than that of the untr-HT group rats during the 7-week period. Figure 1D shows that after 7 weeks of experimentation, the microcirculation blood flow perfusion of the Nifedipin group and the Gra-I group rats significantly increased compared to that of the untr-HT group, with no significant difference between the Nifedipin group and the Gra-I group. Figure 1E shows the instantaneous microcirculation blood flow perfusion of rats in each group after 7 weeks of experimentation. As can be seen from the graph, the instantaneous microcirculation blood flow perfusion of the Nifedipin group and Gra-I group rats was significantly higher than that of the untr-HT group. Therefore, it can be seen that graphene far-infrared irradiation can effectively decrease the blood pressure level of primary hypertensive rats, which is comparable to the effect of Ca^2+^ channel blockers. At the same time, graphene far-infrared irradiation can significantly increase the microcirculation blood flow perfusion of SHRs, reduce peripheral circulation resistance, and also play a role in regulating blood pressure.

### 2.2. Differences in Aorta Morphology and F-Actin Expression in Vascular Smooth Muscle of Rats among All Groups

Figure 2 shows the differences in aortic morphology and structure among the groups of rats. Figure 2A displays the H&E staining results of aortic morphology and structure. The thickness of the arterial wall (Figure 2B) and the number of arterial wall cells (Figure 2D) were analyzed by ImageJ software (win-64). The ratio of the arterial wall thickness to the diameter of the lumen was calculated for each group of rats (Figure 2C). As shown in the figure, the arterial wall thickness of the untr-HT group rats was significantly higher than that of the Control group rats, while the arterial wall thickness of the Gra-I group rats was significantly lower than that of the untr-HT group rats. The number of arterial wall cells in the untr-HT group rats was significantly higher than that in the Control group rats, whereas the number of arterial wall cells in the Gra-I group rats was significantly lower than that in the untr-HT group rats. There was no significant difference in the number of arterial wall cells between the Control group and the Gra-I group rats.

Figure 3 shows the expression of F-actin in the smooth muscle of the aorta in each group of rats. Figure 3A shows the staining of F-actin in the smooth muscle of the aorta using phalloidin. It can be seen from the figure that the positive rate of F-actin in the smooth muscle of the aorta in the untr-HT group was significantly higher than that in the Control group, while the positive rate in the Gra-I group was significantly lower than that in the untr-HT group. We further analyzed the differences in the positive area of F-actin in the smooth muscle of the aorta among the groups (Figure 3B). It can be seen from the figure that the positive area in the untr-HT group was significantly higher than that in the Control group, while the positive area in the Gra-I group was significantly lower than that in the untr-HT group, and there was no significant difference between the Control and Nifedipin groups.

### 2.3. The Expression Levels of eNOS, Ang-II, and ET-1 in Serum of Rats in Each Group

The expression levels of eNOS, Ang-II, and ET-1 in the serum of each group of rats after 7 weeks of the experiment are shown in Figure 4.

As shown in Figure 4A, the expression level of eNOS in the untr-HT group was significantly decreased compared to the Control group. Compared with the untr-HT group and the Nifedipin group, the expression level of eNOS in the Gra-I group was significantly increased. There was no significant difference in the expression level of eNOS between the Nifedipin group and the Gra-I group. Figure 4B shows the expression level of ET-1 in the serum of each group of rats. The expression level of ET-1 in the untr-HT group was significantly increased compared to the Control group, while the expression level of ET-1 in the Gra-I group was significantly decreased compared to the untr-HT group. There was no significant difference in the expression level of ET-1 between the Nifedipin group and the Gra-I group. Figure 4C shows the expression level of Ang-II in the serum of each group of rats. As shown in the figure, the expression level of Ang-II in the untr-HT group was significantly increased compared to the Control group. Compared with the untr-HT group, the expression level of Ang-II in the Gra-I group was significantly decreased, and the expression level of Ang-II in the Nifedipin group was significantly increased. Compared with the Nifedipin group, the expression level of Ang-II in the Gra-I group was significantly decreased.

### 2.4. The Expression Levels of eNOS, α-SMA, and NO in the Aortic Tissues of Each Group of Rats

Figure 5 represents the expression levels of eNOS, α-SMA, and NO in the aortic tissues of rats from each group after 7 weeks of the experiment. Figure 5A,B show the mRNA expression levels of eNOS and α-SMA, respectively. Compared with the Control group, the expression levels of eNOS and α-SMA genes in the untr-HT group were significantly reduced. Compared with the untr-HT group, the expression levels of eNOS and α-SMA genes in the Nifedipin and Gra-I groups were significantly increased. Correspondingly, Figure 5C,D show the protein expression levels of eNOS and α-SMA in the aortic tissues, respectively. Compared with the Control group, the protein expression levels of α-SMA in the untr-HT group were significantly reduced. Compared with the untr-HT group, the expression level of the eNOS protein in the Gra-I group was extremely significantly decreased, while the expression level of the α-SMA protein was extremely significantly increased. There was no significant difference in the expression levels of the α-SMA protein between the Nifedipin group and the untr-HT group.

On this basis, we further compared the differences in NO concentration (Figure 5E) in the arterial tissues of rats from different groups after 7 weeks of the experiment. As shown in Figure 5E, the NO concentration in the arterial tissues of rats in the untr-HT group was significantly lower than that in the Control group, while the NO concentration in the arterial tissues of rats in the Nifedipin and Gra-I groups was significantly higher than that in the untr-HT group, with the NO concentration in the Gra-I group being significantly higher than that in the Nifedipin group. 

## 3. Discussion

Studies have shown that far-infrared radiation can improve blood circulation, regulate blood pressure, and improve endothelial cell function [6], while also potentially reducing the incidence of certain vascular-related diseases [7]. Because the wavelength of far-infrared radiation emitted by graphene material after being electrified falls within the wavelength range easily absorbed by living organisms [16], this study combined the above research background to investigate the alleviating effect of far-infrared radiation generated by electrified graphene material on primary hypertension in rats using the SHR untr-HT. Starting from one week of graphene electric radiation, the changes in blood pressure of each group of rats were continuously monitored. The results showed that within the 7-week period, the blood pressure of primary hypertensive rats irradiated with graphene showed a decreasing trend, indicating that graphene far-infrared radiation has a relieving effect on hypertension. The blood pressure levels of each group of rats fluctuated in various time periods, which we believe is mainly due to stress reactions during the testing process. Based on this phenomenon, we further explored the mechanism by which graphene far-infrared radiation improves primary hypertension.

The decrease in vascular contractility and the increase in intravascular fluid volume are the direct causes of primary hypertension in the body. The factors that can lead to this direct cause are potential causes of primary hypertension. These factors are often coexistent and intertwined in the development of primary hypertension [15]. Among them, the decrease in vascular smooth muscle (VSM) contractility [17,18,19] and microcirculatory dysfunction of blood are two important factors leading to primary hypertension [20]. The contractile function of VSM is regulated by vascular smooth muscle cells (VSMCs), which are the main component cells of the medial layer of the vascular wall and have functions such as proliferation, migration, and secretion of the extracellular matrix. At the same time, due to their high plasticity and phenotypic differentiation ability, VSMCs play an important role in vascular reuntr-HTing [21,22]. Under pathological conditions related to hypertension and vascular injury, highly differentiated VSMCs can dedifferentiate and transition from a contractile to a proliferative phenotype, leading to vascular reuntr-HTing and dysfunction [23,24]. This can result in a significant decrease in the contractile function of cells [25,26], leading to a decline in the VSM’s contractile ability. The above mechanism is one of the main causes that lead to structural arterial sclerosis and induce primary hypertension in the body.

Currently, six phenotypes of VSMCs have been reported, with the contractile phenotype rich in α-smooth muscle actin (α-SMA), which has strong contractile function [25]. The expression level of fibrous actin (F-actin) can reflect the phenotype changes in VSMCs, and when vascular reuntr-HTing and functional abnormalities occur, the expression level of F-actin increases [27]. Based on this, we compared the expression levels of α-SMA and F-actin as well as the morphological structure of the aorta between the Gra-1 group of SHRs treated with graphene and the untr-HT group of untreated SHRs. The results showed that compared to the untr-HT group, the expression of α-SMA in the aorta of the Gra-1 group rats was significantly increased, while the expression of F-actin was significantly decreased. The thickness of the aortic wall was significantly lower in the Gra-1 group than in the untr-HT group, and the number of aortic wall cells was significantly reduced. Therefore, it can be inferred that graphene far-infrared radiation irradiation has a relieving effect on hypertension, which may be achieved by changing the phenotype of VSMCs in the SHR untr-HT and enhancing the contractile function of VSM.

Currently, the NO-sGC-cGMP pathway and RAAS are widely studied pathways that are closely related to VSMC contraction and relaxation [15,28]. The NO-sGC-cGMP pathway originates in the endothelial cells of blood vessels and regulates VSMC contraction and relaxation through a series of signal transductions. The signal transduction mediated by this pathway is considered a potential pathophysiological mechanism for many cardiovascular diseases [29,30,31]. With the discovery of nitric oxide (NO) as one of the endothelium-derived relaxing factors, a large body of evidence suggests that NO produced by nitric oxide synthase (NOS) participates in both the prevention and pathogenesis of human cardiovascular diseases via the NO-sGC-cGMP pathway [29]. Among the factors involved in this pathway, eNOS and some essential cofactors are considered to be of great importance [32]. NO produced by eNOS diffuses into vascular smooth muscle cells and activates the NO-sGC-cGMP pathway, resulting in vasodilation and ultimately relieving blood pressure [16]. In addition, activation of eNOS promotes endothelium-dependent vasodilation, reduces the secretion of ET-1, reduces damage to endothelium-dependent vasodilation [33,34], increases microvascular blood flow, reduces peripheral vascular resistance, and plays a role in relieving blood pressure [20].

Ang II in the RAAS system is a key hormone that can affect almost all organ functions and has both physiological and pathological effects [28]. When subjected to chronic stimulation, the binding of Ang II to receptors in smooth muscle cells can promote VSMC proliferation and hypertrophy [35], which is one of the important factors that affect vascular smooth muscle function in patients with primary hypertension [36].

After 7 weeks of graphene irradiation in the SHRs, compared to the untr-HT group, the expression level of eNOS in the serum of rats in the Gra-I group significantly increased, while the expression levels of ET-1 and Ang-II significantly decreased. The expression levels of eNOS and α-SMA genes in the aortic tissue of the Gra-I group significantly increased, while the expression levels of the eNOS protein decreased significantly and the α-SMA protein increased significantly. The negative correlation between the expression levels of the eNOS gene and protein in the aortic tissue may be due to a series of post-translational mechanisms controlling eNOS expression. The activity of eNOS is affected by S-nitrosylation, acetylation, and c-terminal reductase. We further compared the differences in NO and eNOS protein expressions in the aorta of each group of rats, and the results showed a significant negative correlation. It was inferred that the reasons for this situation mainly include the following two aspects: (1) The expression level of the eNOS protein in the aorta is affected by the concentration of NO. After graphene irradiation, the NO concentration in the aorta of the Gra-I group rats is sufficient, which in turn inhibits the expression of the eNOS protein. (2) After graphene far-infrared radiation, the activity of the eNOS protein in the aorta is enhanced, which produces a multiplier effect. The specific mechanism is still unclear and requires further study.

Since intracellular Ca^2+^ not only directly affects VSMCs [15] but also has an effect when the local concentration of Ca^2+^ changes, Ca^2+^ can activate calmodulin (CaM) to promote electron transfer between structural domains, accelerate NO synthesis [37], and thus affect the NO-sGC-cGMP pathway. Therefore, in this study, the Ca^2+^ blocker nifedipine was used to treat the SHR untr-HT (Nifedipin) and compared with the graphene-irradiated SHR untr-HT. Through experiments, it was found that compared with the Nifedipin group, the Gra-I group SHR untr-HT could effectively improve VSMC phenotype, improve the morphological structure of aortic tissue, enhance the vasomotor function, and at the same time, the concentration of NO in the aortic tissue of the Gra-I group SHR untr-HT was significantly higher than that of the Nifedipin group. Therefore, it can be inferred that for the blood pressure relief effect on the SHR untr-HT, graphene far-infrared irradiation is significantly superior to Ca^2+^ blocker drugs.

Therefore, based on this study, it was found that graphene far-infrared irradiation can effectively change the phenotype of VSMCs and increase microcirculation blood flow by regulating the changes in major cytokines in the NO-sGC-cGMP and RAAS pathways, thereby playing a role in relieving blood pressure and providing a theoretical basis for further clinical research. Since only one irradiation time and one power were used in this study, it is not possible to determine the optimal conditions for graphene irradiation. In future studies, different irradiation times and powers can be selected to further explore the optimal irradiation conditions.

## 4. Materials and Methods

### 4.1. Ethics Statement

All procedures in this study were approved by the Animal Protection and Ethics Committee of Beijing University of Agriculture (Approval No. BUA2022036).

### 4.2. Experimental Animal Selection and Grouping

SPF-level Wistar Kyoto (WKY) rats were used as blank controls, and spontaneously hypertensive rats (SHRs) were used as the untr-HT animals (purchased from Beijing Vital River Laboratory Animal Technology Co., Ltd. (Beijing, China), license number SCXK(Jing)2006-0009). After being transferred to a new SPF animal room, the purchased rats were adaptively fed for 7 days before the experiment. The SHRs were divided into three groups, each containing 5 rats: the untr-HT group (untr-HT), the drug treatment group (Nifedipin), and the graphene irradiation group (Gra-I). Meanwhile, 5 WKY rats were selected as the blank Control group (Control), and the grouping information is shown in Table 1.

### 4.3. Sample Processing

The treatment of each group of rats is shown in Figure 6. The Ca^2+^ channel blocker, nifedipine (Solarbio, SN8410, Beijing, China), was used as the drug for the Nifedipin group at a dose of 10 mg/kg, which was administered orally once a day for 7 consecutive weeks. The SHRs in the Gra-I group were irradiated with a graphene chamber (Appendix A, purchased from Chang Qing Xi Wang Shandong Medical Technology Co., Ltd. (Jinan, Chian)) for 1 h a day, for 7 consecutive weeks. During the experiment, tail artery pressure and blood flow perfusion were measured weekly. After 7 weeks, the rats were anesthetized with 2% pentobarbital sodium solution, and peripheral blood was collected from each group. The aortic tissue of each individual in each group was dissected. The dissected aortic tissue was divided into three parts: one was fixed with paraformaldehyde and stored at room temperature, one was packaged and frozen at −80 °C, and the other was embedded in OCT and rapidly frozen and directly stored at −80 °C. After being left undisturbed at room temperature for 2 h, the peripheral blood was centrifuged horizontally at 4 °C and 3000 r/min for 20 min. Next, the serum was collected and packaged and then frozen at −80 °C.

### 4.4. Non-Invasive Measurement of Tail Artery Blood Pressure and Blood Flow Perfusion

To detect changes in blood pressure in each group of rats, we used a non-invasive tail artery blood pressure meter (ZS-Z, Beijing Zhongshidi Biological Technology Development Co., Ltd. (Beijing, China)) to measure the tail artery pressure of each group of rats every week. First, the rats were induced to enter the restraint device, and when the rats’ status was stable, the measuring ring was slowly placed at about 1/3 of the base of the rat’s tail. The power was turned on and the measurement was started when the waveform on the display screen was stable. The value was recorded, and each rat was measured three times in a row, and the average of the three measurements was used for analysis. The blood flow perfusion of the same area in each group of rats was measured using a laser Doppler blood flow perfusion imager (Peri Cam PSI, Stockholm, Sweden). Before measurement, the rats were anesthetized with a 2% pentobarbital sodium solution, and the measurement was performed after entering deep anesthesia. Each rat was measured three times, each time for 2 min, and the values were recorded. The average of the three measurements was used for statistical analysis.

### 4.5. Enzyme-Linked Immunosorbent Assay (ELISA)

To investigate the changes in the major cytokines in the two main pathways that affect blood pressure changes in the SHRs, we detected the expression levels of eNOS, Ang-II, and ET-1 in serum samples using the ELISA method. After thawing at 4 °C, serum samples were tested for the concentrations of eNOS, Ang-II, and ET-1 using commercial ELISA kits (eNOS from Meimian, MM-0453R1, Nanjing, China; Ang-II from Meimian, MM-0122R1, China; and ET-1 from Meimian, MM-0560R1, China) for each cytokine, with three biological replicates for each sample. The experimental procedures were strictly followed according to the instructions of the kits.

### 4.6. Real-Time Fluorescence Quantitative PCR (RT-PCR)

The total RNA extracted from the aortic tissue was reverse-transcribed into cDNA using the cDNA Synthesis Kit (Tsingke, TSK301, Beijing, China). GAPDH was used as an internal reference gene, and primers (Table 2) were designed using Primer Premier 5.0 software (Premier Biosoft, Palo Alto, CA, USA) and synthesized by Beijing Kaituo Danbai Biotechnology Co., Ltd. (Beijing, China) Real-time fluorescence quantitative PCR (RT-qPCR) was performed on a CFX96 qPCR System (Bio-Rad (Hercules, CA, USA)). The reaction system (20 μL) consisted of 2 μL of cDNA template, 2 μL of primers, 10 μL of PowerUp SYBR Green Master Mix (Applied Biosystems (Waltham, MA, USA), 00791640), and 7 μL of ddH_2_O. The reaction conditions were as follows: pre-denaturation at 94 °C for 5 min, denaturation at 94 °C for 15 s, annealing at 58 °C for 15 s, and extension at 72 °C for 45 s, for 40 cycles. The Ct values were obtained, and the relative expression of the target gene was calculated using the 2^−ΔΔCT^ method.

### 4.7. Western Blot

The Western blot technique was performed to detect the expression levels of eNOS and α-SMA proteins in the aortic tissue. The total protein of the extracted aortic tissue was determined by the BCA assay kit, and an equal amount of protein was subjected to polyacrylamide gel electrophoresis. The protein was then transferred to a PVDF membrane, which was blocked with a 5% non-fat milk solution at room temperature for 1 h. The membrane was incubated overnight at 4 °C with primary antibodies against β-actin (Proteintech, 66009-1-IG, Wuhan, China), eNOS (Proteintech, 27120-1-AP, China), and SMA (Proteintech, 14395-1-AP, China). After washing the PVDF membrane three times with TBST, it was incubated with HRP-labeled goat anti-rat IgG (Beyotime, A0216, Shanghai, China) solution at room temperature for 1.5 h. The membrane was washed and chemiluminescent reagents were added. The chemiluminescent signals were detected and analyzed using the Tanon-5200 chemiluminescent imaging system (Shanghai Tianneng Technology Co., Ltd., Shanghai, China), and the protein band gray values were analyzed using ImageJ software.

### 4.8. Detection of NO Concentration in Aortic Tissue

After thawing the aortic tissue, it was homogenized in a physiological saline solution and a portion of the supernatant was taken. The nitric oxide (NO) concentration in the aortic tissue was determined using a nitric oxide assay kit (Nanjing Jiancheng, A013-2-1, Nanjing, China) according to the manufacturer’s instructions.

### 4.9. Observation of Vascular Tissue Morphology and Smooth Muscle Actin Structure

To observe the vascular tissue morphology in different groups of rats, we embedded the formalin-fixed aortic tissue in paraffin and cut it into 4 μm thick tissue sections. The sections were stained with hematoxylin (Zhongshan Golden Bridge Bio-technology, ZLI-9610, Beijing, China) and eosin (Zhongshan Golden Bridge Bio-technology, ZLI-9613, China) and observed under a microscope for morphological analysis and photography. The cell count, wall thickness, and lumen diameter of the arterial wall were measured using ImageJ software, and the ratio of wall thickness to lumen diameter was calculated.

The frozen tissues embedded in OCT were sliced into 5 μm sections. The sections were then stained with FITC-labeled phalloidin (Solarbio, CA1601, Beijing, China) and mounted with Fluoromount-GTM water-soluble mounting medium. The sealed frozen sections were scanned and photographed under a fluorescence microscope, and the positive area of the sections was analyzed using ImageJ software.

### 4.10. Data Processing and Analysis

The data in this study were analyzed and processed using GraphPad Prism 8.0 software. After 3 technique repeats for each mouse, the average value was taken, and the Mean ± SD was used for statistical analysis of the mean value of each individual technique repeat. One-way ANOVA was used for data analysis, with *p* < 0.05 indicating statistically significant differences and *p* < 0.01 indicating extremely significant differences.

## 5. Conclusions

Graphene far-infrared irradiation can effectively change the phenotype of VSMCs and increase microcirculation blood flow by regulating the major cytokines in the NO-sGC-cGMP and RAAS pathways, thereby playing a role in relieving blood pressure.

## Figures and Tables

**Figure 1 ijms-25-06675-f001:**
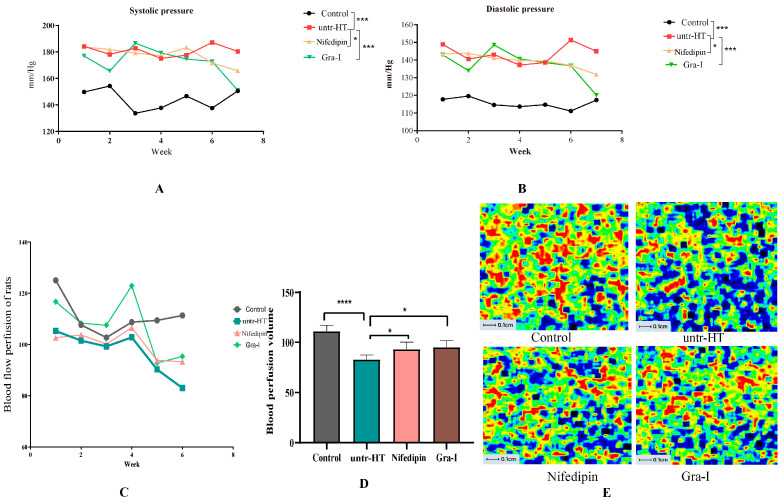
Changes in blood pressure and microcirculation blood flow of rats in each group during a 7-week period. (**A**) Changes in systolic blood pressure of rats in each group over 7 weeks; (**B**) changes in diastolic blood pressure of rats in each group over 7 weeks; (**C**) changes in microcirculation blood flow of rats in each group over 7 weeks; (**D**) differences in microcirculation blood flow between groups after 7 weeks of experiment; and (**E**) instantaneous microcirculation blood flow of rats in each group after 7 weeks of experiment. * Indicates a significant difference between two groups (*p* < 0.05); *** indicates an extremely significant difference between two groups (*p* < 0.01); **** indicates an extremely significant difference between two groups (*p* < 0.01).

**Figure 2 ijms-25-06675-f002:**
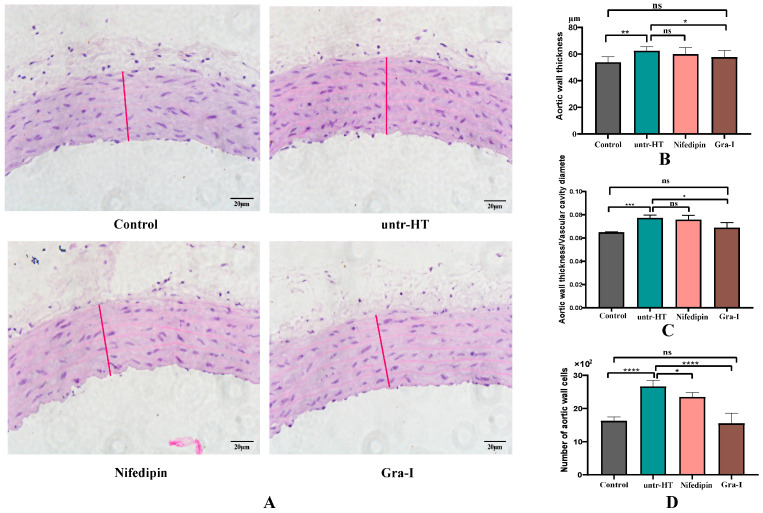
Differences in aortic morphology and structure among groups of rats. (**A**) Morphological structure of the aortic wall in each group of rats; (**B**) differences in aortic wall thickness among groups of rats; (**C**) ratio of aortic wall thickness to lumen diameter in each group of rats; and (**D**) differences in the number of aortic wall cells among groups of rats. * Indicates a significant difference between two groups (*p* < 0.05); ** indicates a significant difference between two groups (*p* < 0.05); *** indicates an extremely significant difference between two groups (*p* < 0.01); **** indicates an extremely significant difference between two groups (*p* < 0.01) and ns indicates no significant difference between two groups.

**Figure 3 ijms-25-06675-f003:**
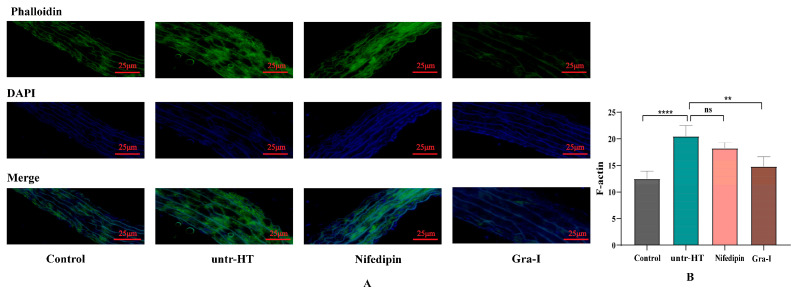
Differences in F-actin expression levels in the aortic smooth muscle of the various rat groups. (**A**) Fluorescence images of F-actin in the aortic smooth muscle of the various rat groups; (**B**) differences in F-actin expression levels in the aortic smooth muscle of the various rat groups. ** Indicates a significant difference between two groups (*p* < 0.05); **** indicates an extremely significant difference between two groups (*p* < 0.01); ns indicates no significant difference between two groups.

**Figure 4 ijms-25-06675-f004:**
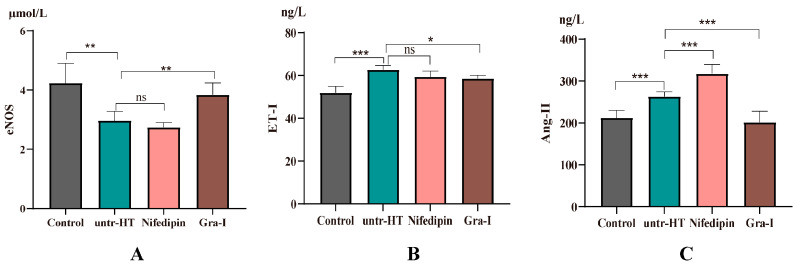
Relative expression levels of eNOS (**A**), ET-1 (**B**), and Ang-II (**C**) in the serum of rats in each group after 7 weeks of experiment. * Indicates a significant difference between two groups (*p* < 0.05); ** Indicates significant difference between two groups (*p* < 0.05); *** indicates an extremely significant difference between two groups (*p* < 0.01); ns indicates no significant difference between two groups.

**Figure 5 ijms-25-06675-f005:**
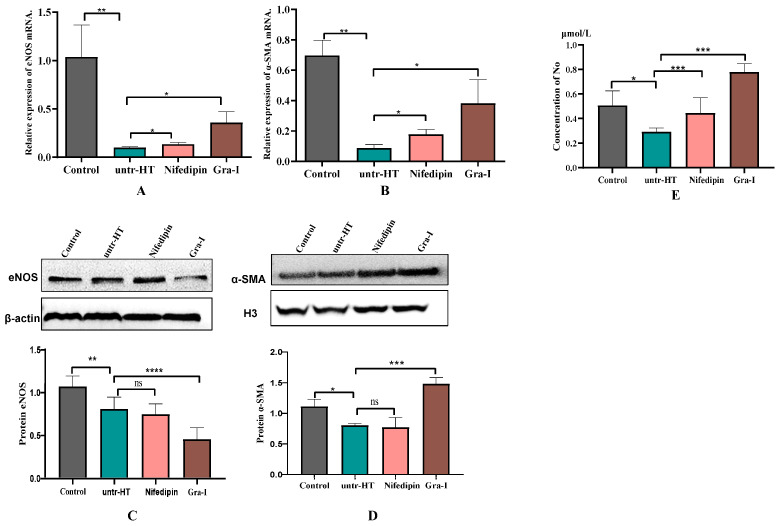
Relative expression levels of eNOS, α-SMA, and NO in the aortic tissues of rats from different groups. (**A**) The expression level of the eNOS gene; (**B**) the expression level of the α-SMA gene; (**C**) the expression level of the eNOS protein; (**D**) the expression level of the α-SMA protein; and (**E**) the concentration of NO in the aortic tissues. * Indicates a significant difference between two groups (*p* < 0.05); ** Indicates significant difference between two groups (*p* < 0.05); *** indicates an extremely significant difference between two groups (*p* < 0.01); **** indicates extremely significant difference between two groups (*p* < 0.01); and ns indicates no significant difference between two groups.

**Figure 6 ijms-25-06675-f006:**
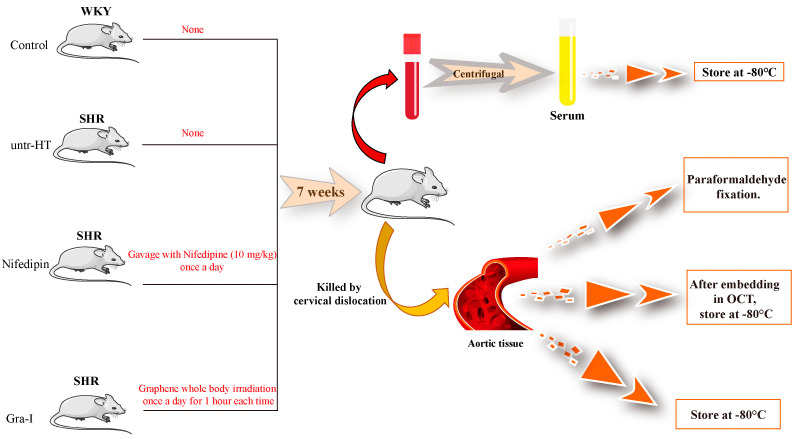
Experimental grouping and sample processing diagram.

**Table 1 ijms-25-06675-t001:** Animal grouping information.

Group	Rat Type	Number
Control	WKY	5
untr-HT	SHR	5
Nifedipin	SHR	5
Gra-I	SHR	5

**Table 2 ijms-25-06675-t002:** Real-time fluorescence quantitative PCR primer sequences.

Gene	Primer Name	Primer Sequence
*GAPDH*	GAPDH-F	TGAAGCTCATTTCCTGGTATGAC
	GAPDH-R	GGCCTCTCTCTTGCTCTCAGTA
*eNOS*	eNOS-F	GGCTGAGTACCCAAGCTGAG
	eNOS-R	ATTGTGGCTCGGGTGGATTT
*α-SMA*	α-SMA-F	GCCGAGATCTCACCGACTAC
	α-SMA-R	ACGATCTCACGCTCAGCAGTA

## Data Availability

All data and information related to this study can be obtained through the corresponding author.

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
