# Peer review of "Graphene Far-Infrared Irradiation Can Effectively Relieve the Blood Pressure Level of Rat Untr-HT in Primary Hypertension"

_ijms, 2024, doi:10.3390/ijms25126675_

Round 1

Reviewer 1 Report

Comments and Suggestions for Authors

Dr Lu and colleagues have submitted a manuscript on graphene far-infrared irradiation on hypertensive rats. They have shown that hypertensive rats irradiated with far-infrared radiation from electrified graphene have lower blood pressure levels and changes in cytokines. 

Over all, the manuscript is very well written and very interesting. I do have some comments, however. 

- Beginning of the results section: I am aware that the journal requests the methods only at the end of the manuscript. This way, it is very difficult to understand, what the differences between the groups are. It might be helpful to very briefly describe the different rat types. Another helpful thing might be to rename the groups: Controls remain controls, Model e.g. "untr-HT", Pos-C e.g. "Nifedipin", Gra-I remain Gra-I. 

- What is your explanation for the initial increase in BP?

- BP: Gra-I have more or less the same or higher BP during most measurements, except the last one. How do you make sure, that there is truly a decrease in BP and that this is not just a false measurement? You only have 5 rats per group, so a single very low or high measurement can change the meann (did you use mean or median? I can't find this in the methods).

- Perfusion: why is the perfusion decreasing in the control-group? Why is there such a sharp increase and then decrease in perfusion in the Gra-I group?

- Figures: try to use the same colours for the same groups in all figures. Why are some bars filled and some not? 

- Figure 2: wall thickness measurement appears to be non-perpendicular in the Gra-I aorta. 

- Figure 4: why do you show different comparisons in the different panels (also Figure 1, 2, 3). Use the same comparisons in all figures or give a clear explanation, why you compare only these particular groups. 

- Methods: I don't understand from you description how the irradiation works. How big is the chamber? What happens to the rat during the irradiation, is it anesthetized?

- as mentioned above, write how numbers are reported and what the whiskers in figures mean.  

- as a final and more general question: do you think that the effect which you show is dependent on graphene or is it the far-infrared irradiation in general?

Comments on the Quality of English Language

There are some minor errors, particularly incomplete sentences for example at the beginning of the results section and some grammatical errors in the abstract. Over all, the quality of English Language is good. 

Reviewer 2 Report

Comments and Suggestions for Authors

Dear Authors, 

This is a very interesting article regarding the use of far infrared irradiation in modulating blood pressure in the spontaneously hypertensive rat model. The study is well designed and the results are innovative, but you submitted the article to the special issue "Advances in Biocompatible Materials for Dental Applications".

I believe the you should submit this article to an issue compatible with the topic covered.

Comments on the Quality of English Language

The paper requires revision in English because it has a number of grammatical and hyphenation inaccuracies

Round 2

Reviewer 2 Report

Comments and Suggestions for Authors

Dear Authors,

This is a very interesting article regarding the use of far infrared irradiation in modulating blood pressure in the spontaneously hypertensive rat model. The study is well designed and the results are innovative, as already stated in the previous review.

Comments on the Quality of English Language

Minor editing of English language required